# Physiological and Biochemical Evaluation of Different Types of Recovery in National Level Paralympic Powerlifting

**DOI:** 10.3390/ijerph18105155

**Published:** 2021-05-13

**Authors:** Wélia Yasmin Horacio dos Santos, Felipe J. Aidar, Dihogo Gama de Matos, Roland Van den Tillaar, Anderson Carlos Marçal, Lázaro Fernandes Lobo, Lucas Soares Marcucci-Barbosa, Saulo da Cunha Machado, Paulo Francisco de Almeida-Neto, Nuno Domingos Garrido, Victor Machado Reis, Érica Leandro Marciano Vieira, Breno Guilherme de Araújo Tinoco Cabral, José Vilaça-Alves, Albená Nunes-Silva, Walderi Monteiro da Silva Júnior

**Affiliations:** 1Graduate Program in Physical Education, Federal University of Sergipe (UFS), São Cristovão 49100-000, Sergipe, Brazil; weliaa@hotmail.com (W.Y.H.d.S.); acmarcal@yahoo.com.br (A.C.M.); walderim@yahoo.com.br (W.M.d.S.J.); 2Department of Physical Education, Federal University of Sergipe (UFS), São Cristovão 49100-000, Sergipe, Brazil; 3Group of Studies and Research of Performance, Sport, Health and Paralympic Sports—GEPEPS, The Federal University of Sergipe (UFS), São Cristovão 49100-000, Sergipe, Brazil; dihogogmc@hotmail.com; 4Graduate Program of Physiological Science, Federal University of Sergipe—UFS, São Cristovão 49100-000, Sergipe, Brazil; 5Cardiovascular & Physiology of Exercise Laboratory, University of Manitoba, Winnipeg, MB R3T 2N2, Canada; 6Department of Sport Sciences and Physical Education, Nord University, 7600 Levanger, Norway; roland.v.tillaar@nord.no; 7Laboratory of Inflammation and Exercise Immunology, Sports Center, Federal University of Ouro Preto, Ouro Preto 35400-000, Minas Gerais, Brazil; lobocav@hotmail.com (L.F.L.); lucasmarcucci@gmail.com (L.S.M.-B.); albenanunes@hotmail.com (A.N.-S.); 8Graduate Program in Health Science, Federal University of Sergipe (UFS), São Cristovão 49100-000, Sergipe, Brazil; saulo0407@gmail.com; 9Department of Physical Education, Federal University of Rio Grande do Norte, Lagoa Nova 59078-970, Natal, Brazil; paulo220911@hotmail.com (P.F.d.A.-N.); brenotcabral@gmail.com (B.G.d.A.T.C.); 10Research Center in Sports Sciences, Health Sciences and Human Development (CIDESD), Trás os Montes and Alto Douro University, 5000-801 Vila Real, Portugal; ngarrido@utad.pt (N.D.G.); victormachadoreis@gmail.com (V.M.R.); josevilaca@utad.pt (J.V.-A.); 11Interdisciplinary Medical Research Laboratory (LIIM), Faculty of Medicine, Federal University of Minas Gerais, Belo Horizonte 30130-100, Minas Gerais, Brazil; ericalmvieira@gmail.com; 12Department of Physical Therapy, Universitary Hospital, Federal University of Sergipe (UFS), São Cristovão 49100-000, Sergipe, Brazil

**Keywords:** Paralympic powerlifting, plasma cytokines, recovery, cold water immersion, dry needling

## Abstract

Background: Recovery from training is vital as it ensures training and performance to continue at high intensities and longer durations to stimulate the body and cause further adaptations. Objective: To evaluate different methods of post-workout recovery in Paralympic powerlifting athletes. Methods: Twelve male athletes participated (25.4 ± 3.3 years; 70.3 ± 12.1 kg). The presence of muscle edema, pain threshold, plasma cytokines, and performance measurement were evaluated five times. The recovery methods used in this study were passive recovery (PR), dry needling (DN), and cold-water immersion (CWI). Results: The data analysis showed that the maximal force decreased compared to the pretest value at 15 min and 2 h. The results also revealed that CWI and DN increased Interleukin 2 (IL-2) levels from 24 to 48 h more than that from 2 h to 24 h. After DN, muscle thickness did not increase significantly in any of the muscles, and after 2 h, muscle thickness decreased significantly again in the major pectoralis muscle. After CWI, pain pressure stabilized after 15 min and increased significantly again after 2 h for acromial pectoralis. Conclusion: The strength training sessions generate several changes in metabolism and different recovery methods contribute differently to maintain homeostasis in Paralympic powerlifting athletes.

## 1. Introduction

Post-exercise recovery has been the focus of intensive investigation in the scientific community because of its importance in current physical training programs at different performance levels, especially at the high levels of performance at which the athletes train more than once a day [1,2]. The adequate post-exercise recovery strategy is an essential aspect of any physical conditioning program [3,4,5] and consists of restoring the body’s systems to their basal conditions [4].

Monitoring of post-exercise recovery is important to ensure better quality in all subsequent training sessions. The inadequacy between volume and intensity in training sessions and rest periods can go beyond the individual limit of athletes, resulting in unnecessary wear and tear [6]. Therefore, it is necessary to give equal importance to both training and recovery before submitting the athlete to a new training session or competition to maintain a healthy balance, avoid performance restriction, and decrease injury risk [4].

Several recovery methods have been proposed, such as stretching, massage, cryotherapy, relaxation techniques, thermotherapy, and physiotherapy techniques [7]. These recovery methods must work daily and allow for quick access to selected markers, providing immediate feedback of the subject’s psychophysical condition, leading to interventions in training loads and performance improvements [7].

Cold water immersion (CWI) is commonly used to relieve pain, particularly inflammatory diseases, injuries, and overuse [8]. Furthermore, recovery in post-workout CWI in athletes has been widely used, with positive effects on physiological aspects, the inflammatory process, and metabolic and nerve transmission [9,10,11]. Cooling the body to accelerate recovery decreases muscle damage and inflammation markers [8]. CWI involves submersing a part of or the whole body, except the head, in a cold-water bath (11 to 15 °C) for 11 to 15 min [12,13]. However, although athletes have widely used CWI, this type of recovery is not yet used in Paralympic athletes due to the fact that these athletes have difficulties in being inserted/removed from the place of the recovery process.

In addition, the use of dry needling (DN) in recovery has been proposed to reduce muscle pain [14,15]. This technique acts through a local inflammatory process by increasing blood supply to the injured area. However, there are no reports of its application to alleviate late muscle pain symptoms [14,15]. Dry needling is a relatively new technique for recovery, which already shows good results in recovery overuse, which is common in sports [16].

The literature proposes some recovery methods for athletes, including CWI [17], non-steroidal anti-inflammatory drugs (NSAIDs) (i.e., ibuprofen) [18], and dietary supplements [19,20]. Furthermore, there is a paucity of scientific information on the effects of post-exercise recovery methods on Paralympic powerlifting athletes. Therefore, we aim to evaluate the effects of different post-training recovery methods on mechanical, biochemical, and pain scales in Paralympic powerlifting athletes. Because of this lack of consensus on post-exercise recovery and the difficulty Paralympic athletes have performing a recovery in immersion in cold water, it was hypothesized that DN could be effective in post-exercise recovery in Paralympic athletes. In addition, other recovery techniques have been shown to be just as effective for athletes, which could be tested for Paralympic athletes.

## 2. Materials and Methods

The study was conducted over 4 weeks; in the first week, the athletes were familiarized with the tests and in the second to fourth weeks, different recovery methods were evaluated. The recovery methods evaluated included passive recovery (PR), dry needling (DN), and cryotherapy recovery (Figure 1).

### 2.1. Sample

This was a cross-over study involving 12 male Paralympic powerlifting athletes (age: 25.4 ± 3.3 years; body mass: 70.3 ± 12.2 kg; 1 repetition maximum (RM): 117.4 ± 23.4 kg; 1 RM/body mass: 1.67 ± 0.28). Those who were clinically stable and had been practicing the sport for at least 12 months (2.5 ± 0.2 years), eligible for Paralympic powerlifting, were included. Inclusion criteria also involved those who had participated in an official competition in the past 6 months, were eligible for the sport in accordance with the rules of the International Paralympic Committee (IPC) [21], had been training uninterruptedly for at least 12 months, and were ranked among the top 10 of their respective categories at the national level. The exclusion criteria were not participating in any of the evaluations and making use of any type of legal or illegal ergogenic.

All participants were national-level athletes; two national champions, two runners-up, and two third places. The rest were classified in the top 10 of their respective categories, according to the criteria of the International Paralympic Committee [21]. Regarding disability types, four athletes presented spinal cord injury due to trauma, with injury below the eighth thoracic vertebra; two with sequelae due to polio; four with lower limb malformation (arthrogryposis); and two with cerebral palsy.

This study was approved by the Human Research Ethics Committee of the Federal University of Sergipe (UFS), CAEE: 79909917.0.0000.55.46 (technical advice: 2,637,882, date of approval: 7 May 2018). The procedures adopted followed the standards of ethics in research with humans per Resolution No. 466, of 12 December 2012, which are the guidelines and regulatory standards for research involving humans, in accordance with the ethical principles contained in the Declaration of Helsinki (1964, restated in 1975, 1983, 1989, 1996, 2000, 2008, and 2013) of the “World Medical Association.”

### 2.2. Procedures

One familiarization session in the 2 weeks and at least 72 h before the intervention was conducted. The 1 RM test was performed, in which, initially, the subjects performed the trials with a weight that they estimated could be lifted only once, using the maximum effort. Weight was then added until the maximum load that could be lifted with one repetition was reached. If the participant could not perform a single repetition, 2.4–2.5% of the load used in the test was subtracted [22]. A 3- to 5-min rest was provided between attempts. The minimum interval between tests and the beginning and end of the training session was 10 min.

On the intervention days, the athletes performed a warm-up for the upper limbs, using three exercises (abduction of the shoulders with dumbbells, elbow extension in the pulley, and rotation of the shoulders with dumbbells), with three sets of 10 to 20 RM in approximately 10 min [21]. Subsequently, a specific warm-up was performed on the bench press itself, with 30% of the load for 1 RM, in which 10 slow repetitions (3:1 s: eccentric × concentric) and 10 fast repetitions (1:1 s), controlled with a Willner metronome (Willner, Isny, Germany), were performed to induce muscle damage. During the test, the athletes were verbally encouraged to perform the movement with their best performance [23].

The bench press movement was used, and five sets of five repetitions with a load of 120% in the eccentric phase and 80% in the concentric phase of 1 RM and three additional series of five repetitions, with emphasis on acceleration with a 40% load of 1 RM, were performed. In the five initial repetitions, an extra load was placed in the eccentric phase, which was removed in the concentric phase, making up 120% of 1 RM in the eccentric phase and 80% in the concentric phase. In the three additional repetitions, only 80% of 1 RM was used, while, in the concentrated phase, an evaluator helped the athletes complete three repetitions. Technical training involved the use of loads previously determined for each individual in the maximum repetition test (1 RM). Between the exercise blocks, a 3-min interval for rest was provided.

After the exercise-induced muscle damage, one of the recovery methods was applied, and 10 min of rest were provided without any specific protocol [24]. The athletes remained seated before performing one of the recovery methods. The recovery methods were defined by lot, where 1/3 of the athletes did each recovery method in turn in three sessions until they completed the three recovery methods.

A trained physiotherapist performed the DN technique and explained how it works and how sensations, such as “pinching”, can be perceived during the application due to the stimulus and muscle response. The application site was sterilized using cotton soaked in 70% alcohol, then the individual was placed in the supine position when directing the pectoralis major (between the third and fourth intercostal space under the midpoint of the clavicle, anterior deltoid (upper, anterior, and 1/3 lateral the clavicle to the deltoid tuberosity of the humerus), and in the lateral decubitus when directing the brachial triceps (60% distal between the lateral epicondyle of the humerus and the scapular acromial process), where the needle applications were performed by means of stainless and sterile monofilament (0.25/40 mm), perpendicular to the muscles and held in place for 5 min, not being manipulated or stimulated during the execution of the in situ technique.

According to the criteria established by the study, this technique offers minimal risks to individuals; however, in individuals with a low pain threshold, this stimulus can be painful [25].

In the CWI intervention, the athletes stayed immersed for 15 min in cold water up to their neck (11.0 °C up to 15.0 °C), following the protocol defined by Machado et al. [12] and Vieira et al. [13].

In the passive recovery method, the athletes remained seated for 15 min without any activities. The intervention order was carried out by drawing lots, where 1/3 made a recovery in each method.

### 2.3. Instruments/Measurements

Measurements were taken before and immediately after each recovery method (±15 min after the exercise-induced muscle damage) and after 2, 24, and 48 h (Figure 1). The pressure pain threshold (PPT) was assessed using a digital pressure algometer (Impac^®^, probe with an area of 1.0 cm^2^, São Paulo, SP, Brazil). The instrument was placed perpendicular to the pectoralis major, anterior deltoid, and triceps on each muscle’s motor point, following the anatomical references used in the ultrasound measurements [26].

The evaluator placed continuous pressure on the musculature, and the athletes were instructed to say “stop” when the sensation of pressure became painful. Three measurements were taken for each muscle, and each muscle´s average was recorded [27].

### 2.4. Ultrasound

To measure muscle thickness after exercise, an ultrasound device (Medison Sonoace^®^ brand, General Electric, Boston, MA, USA) was applied to the respective muscles: pectoral and deltoid [26].

Before the tests, each musculature’s measuring points were marked with a felt-tip pen for better identification of the structures. The individuals remained with their muscles relaxed in order not to generate interference in the measurement. The head of the device was used to make a 5-MHz scan with a linear-type transducer in the measurement sites without pressing the tissue, identifying the adipose tissue´s subcutaneous connections and the muscle-bone interface, for which distances were measured, corresponding to the muscle thickness. The device provided the measurement through which the ultrasound images were frozen and the measurements were extracted [28].

### 2.5. Blood Biochemical Indicators

The blood of the individuals participating in the study was collected by qualified health professionals, respecting biosafety techniques, such as the use of gloves and sterile and disposable material. About 30 mL of blood was collected using tubes with heparin, EDTA, and clot activator and sterile and disposable material (Vacutainer, Grand Island, NY, USA). According to the manufacturer’s instructions, the serum blood concentrations of plasma cytokine were measured using the cytometric bead array (CBA) method, with Human T helper (Th) (Th1/Th2 Cytokine Kit II: brand BD Biosciences, San Jose, CA, USA). These kits were used to quantify inflammatory protein cytokines interleukin (IL), IL-2, and IL-4 and interferon gama (IFN-γ). The plasma samples were incubated, with the capture microspheres covered by specific antibodies for the respective cytokines and chemokines, as well as the proteins of the standard curve. The color reagent (Ficoeritrina: PE) was added to the samples before being incubated for 3 h. After incubation, the samples were washed (Wash buffer^®^) and centrifuged (1100 rpm, 200× *g* per 5.0 min, room temperature). The supernatant was discarded, and the precipitate containing the microspheres was resuspended with 300 µL of Wash buffer.

The samples were analyzed using the BD FACS Canto II flow cytometer (Becton & Dickinson, San Jose, CA, USA). The results were analyzed using the FCAP software (BD Bioscience, San Jose, CA, USA) and are represented in pg/mL. The blood was stored in a −80 °C freezer.

IL-2 has a pro-inflammatory action and helps in the proliferation of T and B lymphocytes, inducing the production of IFN-γ [29]. IFN-γ has a pro-inflammatory action, activating Macrophage to produce toxic radicals and TNF-α production, which in turn has a pro-inflammatory action, inducing IL-1 shock protein and apoptosis [29,30,31]. IL-4 has an anti-inflammatory action, inhibiting the production of IL-1 α/β, tumor necrosis factor alpha (TNF-α), and IL-6, and inducing Th0–Th2 differentiation and proliferation and differentiation of B lymphocytes [27].

### 2.6. Maximum Isometric Force

The maximum isometric force (MIF) was determined using a Musclelab force sensor (Model PFMA 3010e MuscleLab System; Ergotest, Langesund, Norway) attached to the adapted Bench, employing Spider HMS Simond carabiners (Chamonix, France) with a 21 KN burst load, approved for climbing by the Union Internationale DES Associations d’Alpinisme (UIAA). A steel chain with a breaking load of 2300 kg was used to fix the bench´s force sensor. The perpendicular distance between the force sensor and the joint center was determined and used to calculate joint torques, adapted from the procedure performed by Fraga et al. [18] and Sampaio et al. [19].

The isometric force was determined by a force sensor attached to an inextensible cable and the adapted bench press. Participants were instructed to perform a single maximum movement, looking for elbow extension (90°) (“as fast as possible”) and to relax.

### 2.7. Statistics

For the descriptive analysis, central tendency measures, average (X) ± standard deviation (SD), were used. The normality of the variables was tested using the Shapiro–Wilk test. Because significant differences were found at the baseline between the different recovery methods, a 3 (recovery method) × 4 (test time: pre-15 min, 15 min to 2 h, 2–24 h, and 24–48 h) analysis of variance (ANOVA test) with repeated measures on each variable with the baseline data as a covariate was performed, with Holm–Bonferroni’s post-hoc test. The effect size was determined by the values of “partial eta squared” (η^2^p), considering values of low effect (≤0.05), medium effect (0.05–0.25), high effect (0.25–0.50), and very high effect (>0.50) [32]. The percentage of the variation coefficient (CV%) was calculated for the variables using the formula: CV% = (standard deviation (SD)/mean) × 100. To measure the reliability between the measurements by different methods (Passive, DN, and CW)], we calculated the intraclass correlation coefficient (ICC), whose magnitudes were determined [32]: absence: <0; bad: 0–0.19; weak: 0.20–0.39; moderate: 0.30–0.59; substantial: 0.60–0.79; almost complete: ≥0.80. Statistical analyses were performed using the Statistical Package for the Social Science (SPSS) version 25.0 software (IBM, North Castle, New York, NY, USA). The level of significance was set at *p* < 0.05.

## 3. Results

A significant effect of the recovery method (F = 3.6, *p* = 0.046; η^2^p = 0.25), test occasion (F = 6.7, *p* = 0.001; η^2^p = 0.38), and method × test interaction (F = 7.1, *p* = 0.002; η^2^p = 0.39) were found for MIF. Post-hoc comparison revealed that maximal force decreased from pretest to straight after implementation of the different recovery methods. Both PR and DN increased to the baseline levels of MIF from 2 to 24 h, whereas CWI increased MIF from 2 to 24, but reached 20% more force after 24 h than at the baseline level (Figure 2).

In relation to Figure 2, the %CV for MIF were: passive recovery: pre-test = 37.7%; after 15 min: 26.2%; after 24 h: 24.6%; after 48 h: 30.0%. In DN recovery: pre-test = 33.3%; after 15 min: 22.5%; after 24 h: 24.7%; after 48 h: 23.1%. With regard to the CWI recovery: pre-test = 31.1%; after 15 min: 18.8%; after 24 h: 16.5%; after 48 h: 29.2%. In addition, for MIF, the ICC among the recovery methods (passive, DN, and CWI) was: pre-test: ICC = 0.95 (95% CI: 0.87; 0.99, *p* < 0.0001); after 15 min: ICC = 0.76 (95% CI: 0.43; 0.94, *p* < 0.0001); after 24 h: ICC = 0.47 (95% CI: 0.04; 0.84, *p* = 0.01); after 48 h: ICC = 0.40 (95% CI: −0.02; 0.80, *p* = 0.03).

Only a significant effect of the recovery method (F = 7.1, *p* < 0.006; η^2^p = 0.47) and over moments (F = 6.0, *p* < 0.003; η^2^p = 0.43) in biochemical blood indicators was found in IL-2. Post-hoc comparison revealed that CWI and DN increased IL-2 levels from 24 to 48 h more than that from 2 to 24 h, whereas the other two did not change significantly over time (Figure 3).

In relation to Figure 3, for passive recovery, the %CV of the variables were: IFN-γ: pre-test = 31.8%; after 15 min: 26.2%; after 2 h: 30.6%; after 24 h: 16.6%; after 48 h: 22.7%. IL-4: pre-test = 21.1%; after 15 min: 20.2%; after 2 h: 10.9%; after 24 h: 14.9%; after 48 h: 12.4%. IL2: pre-test = 22.4%; after 15 min: 18.7%; after 2 h: 17.2%; after 24 h: 14.7%; after 48 h: 13.4%.

For DN recovery, the %CV of the variables were: IFN-γ: pre-test = 29.7%; after 15 min: 31.5%; after 2 h: 29.5%; after 24 h: 27.4%; after 48 h: 24.8%. IL-4: pre-test = 20.5%; after 15 min: 23.0%; after 2 h: 17.1%; after 24 h: 15.5%; after 48 h: 18.6%. IL2: pre-test = 12.7%; after 15 min: 18.8%; after 2 h: 17.0%; after 24 h: 16.9%; after 48 h: 16.3%.

For the CWI recovery, the %CV of the variables were: IFN-γ: pre-test = 25.5%; after 15 min: 33.9%; after 2 h: 28.1%; after 24 h: 26.0%; after 48 h: 24.8%. IL-4: pre-test = 27.0%; after 15 min: 22.5%; after 2 h: 23.2%; after 24 h: 32.2%; after 48 h: 20.1%. IL2: pre-test = 20.5%; after 15 min: 17.2%; after 2 h: 18.3%; after 24 h: 26.1%; after 48 h: 21.0%.

In Figure 3, the ICC among the recovery methods (passive, DN, and CWI) for cytokines: IFN-γ: pre-test: ICC = 0.58 (95% CI: 0.17; 0.88, *p* = 0.002); after 15 min: ICC = 0.74 (95% CI: 0.39; 0.93, *p* = 0.0001); after 2 h: ICC = 0.56 (95% CI: 0.15; 0.87, *p* = 0.003); after 24 h: ICC = 0.18 (95% CI: −0.19; 0.68, *p* = 0.1); after 48 h: ICC = 0.20 (95% CI: −0.17; 0.70, *p* = 0.1). IL−4: pretest: ICC = 0.16 (95% CI: −0.20; 0.67, *p* = 0.2); after 15 min: ICC = 0.31 (95% CI: −0.06; 0.78, *p* = 0.05); after 2 h: ICC = 0.14 (95% CI: −0.21; 0.66, *p* = 0.2); after 24 h: ICC = 0.58 (95% CI: 0.16; 0.88, *p* = 0.003); after 48 h: ICC = 0.18 (95% CI: −0.18; 0.69, *p* = 0.1). IL2: pretest: CCI = 0.32 (95% CI: −0.09; 0.76, *p* = 0.06); after 15 min: ICC = 0.78 (95% CI: 0.48; 0.94, *p* < 0.0001); after 2 h: ICC = 0.81 (95% CI: 0.52; 0.95, *p* < 0.0001); after 24 h: ICC = 0.74 (95% CI: 0.40; 0.93, *p* < 0.0001); after 48 h: ICC = 0.32 (95% CI: −0.09; 0.76, *p* = 0.06).

A significant effect of recovery method (F = 8.9, *p* ≤ 0.002; η^2^p = 0.52), test occasion (F = 5.8, *p* ≤ 0.001; η^2^p = 0.42), and interaction effect (F = 7.1, *p* < 0.001; η^2^p = 0.47) on muscle thickness was noted. Post-hoc comparisons revealed that muscle thickness increased mostly after PR in all muscles and remained elevated, whereas, after cryotherapy, only muscle thickness was significantly higher after 15 min and 2 h. After DN, muscle thickness did not increase significantly in any of the muscles and, after 2 h muscle thickness, decreased significantly again in the major pectoralis muscle (Figure 4).

In relation to Figure 4, for passive recovery, the %CV of the variables were: external chest: pre-test = 7.9%; after 15 min: 6.9%; after 2 h: 21.6%; after 24 h: 31.5%; after 48 h: 34.2%. Clavicular pectoralis: pre-test = 7.0%; after 15 min: 9.5%; after 2 h: 28.7%; after 24 h: 29.2%; after 48 h: 25.9%. Triceps: pre-test = 8.2%; after 15 min: 12.2%; after 2 h: 31.5%; after 24 h: 38.6%; after 48 h: 24.9%. Deltoid: pre-test = 11.2%; after 15 min: 19.3%; after 2 h: 17.3%; after 24 h: 23.6%; after 48 h: 24.6%.

For the DN recovery, the %CV of the variables were: external chest: pre-test = 23.7%; after 15 min: 17.6%; after 2 h: 13.9%; after 24 h: 29.5%; after 48 h: 22.3%. Clavicular pectoralis: pre-test = 18.7%; after 15 min: 14.5%; after 2 h: 13.5%; after 24 h: 27.1%; after 48 h: 17.9%. Triceps: pre-test = 14.9%; after 15 min: 24.2%; after 2 h: 17.9%; after 24 h: 23.2%; after 48 h: 24.7%. Deltoid: pre-test = 11.1%; after 15 min: 20.0%; after 2 h: 3.44%; after 24 h: 16.8%; after 48 h: 23.2%.

For the CWI recovery, the %CV of the variables were: external chest: pre-test = 13.8%; after 15 min: 17.3%; after 2 h: 13.8%; after 24 h: 22.1%; after 48 h: 18.2%. Clavicular pectoralis: pre-test = 11.9%; after 15 min: 24.0%; after 2 h: 15.8%; after 24 h: 19.7%; after 48 h: 18.1%. Triceps: pre-test = 12.0%; after 15 min: 20.0%; after 2 h: 16.7%; after 24 h: 14.9%; after 48 h: 20.6%. Deltoid: pre-test = 12.3%; after 15 min: 9.1%; after 2 h: 7.8%; after 24 h: 16.7%; after 48 h: 19.9%.

The ICC between the recovery methods (passive, DN, and CWI) for each muscle was: external chest: pre-test: ICC = 0.33 (95% CI: −0.07; 0.77, *p* = 0.05); after 15 min: CCI = 0.06 (95% CI: −0.26; 0.60, *p* = 0.3); after 2 h: CCI = 0.05 (95% CI: −0.26; 0.59, *p* = 0.3); after 24 h: CCI = −0.04 (95% CI: −0.32; 0.49, *p* = 0.5); after 48 h: ICC = 0.52 (95% CI: 0.09; 0.85, *p* = 0.007). Clavicular pectoralis: pre-test: ICC = 0.19; (95% CI: −0.18; 0.69, *p* = 0.1); after 15 min: ICC = −0.07 (95% CI: −0.33; 0.46, *p* = 0.5); after 2 h: ICC = 0.12 (95% CI: −0.22; 0.64, *p* = 0.2); after 24 h: ICC = 0.04 (95% CI: −0.27; 0.58, *p* = 0.3); after 48 h: ICC = 0.30 (95% CI: −0.10; 0.76, *p* = 0.07). Triceps: pretest = ICC = 0.11 (95% CI: 0.23; 0.64, *p* = 0.2); after 15 min: ICC = 0.24 (95% CI: −0.14; 0.72, *p* = 0.1); after 2 h: ICC = 0.22 (95% CI: −0.16; 0.71, *p* = 0.1); after 24 h: ICC = 0.02 (95% CI: −0.28; 0.56, *p* = 0.4); after 48 h: ICC = 0.40 (95% CI: −0.01; 0.81, *p* = 0.03). Deltoid: pre-test: ICC = 0.70 (95% CI: 0.34; 0.92, *p* = 0.0002); after 15 min: ICC = 0.03 (95% CI: −0.27; 0.57, *p* = 0.3); after 2 h: ICC = 0.29 (95% CI: −0.11; 0.75, *p* = 0.08); after 24 h: ICC = 0.00 (95% CI: −0.29; 0.54, *p* = 0.4); after 48 h: ICC = 0.236 (95% CI: −0.15; 0.72, *p* = 0.1).

Pain pressure threshold was significantly affected by the recovery method for the pectoralis muscles (F = 8.4, *p* ≤ 0.003; η^2^p = 0.51), but not for the deltoid muscle (F = 2.9, *p* = 0.085; η^2^p = 0.26). A significant effect of time was found (F = 5.3, *p* < 0.002; η^2^p = 0.40) for all muscles, together with a significant interaction effect (F = 5.2, *p* ≤ 0.005; η^2^p = 0.39) on pain pressure threshold. Post-hoc comparisons revealed that pain pressure threshold increased significantly immediately after all implementations of recovery methods (15 min). However, the pain pressure threshold continued to decrease for all muscles, with the lowest measurement being noted 24 h after PR, after which it increased again. In DN, a similar development to PR was noted, but the decrease in pain pressure threshold decreased, in general, less than that after PR. After CWI, pain pressure stabilized after 15 min and significantly increased again after 2 h for the pectoralis sternal part (Figure 5).

In relation to Figure 5, for passive recovery, the %CV of the variables were: external chest: pre-test = 11.7%; after 15 min: 5.7%; after 2 h: 11.9%; after 24 h: 6.0%; after 48 h: 3.0%. Clavicular pectoral: pre-test = 9.8%; after 15 min: 11.3%; after 2 h: 4.5%; after 24 h: 5.9%; after 48 h: 8.4%. Triceps: pre-test = 13.1%; after 15 min: 10.2%; after 2 h: 9.0%; after 24 h: 6.8%; after 48 h: 6.7%. Deltoid: pre-test = 6.3%; after 15 min: 7.6%; after 2 h: 4.8%; after 24 h 4.1%; after 48 h: 4.7%.

For DN recovery, the %CV of the variables were: external chest: pre-test = 8.6%; after 15 min: 9.5%; after 2 h: 68%; after 24 h: 7.5%; after 48 h: 14.4%. Clavicular pectoralis: pre-test = 8.4%; after 15 min: 11.9%; after 2 h: 12.6%; after 24 h: 4.2%; after 48 h: 9.5%. Triceps: pre-test = 21.4%; after 15 min: 10.9%; after 2 h: 13.5%; after 24 h: 11.8%; after 48 h: 9.1%. Deltoid: pre-test = 6.4; after 15 min: 7.8; after 2 h: 7.5%; after 24 h 4.9%; after 48 h: 9.0%.

For the CWI recovery, the %CV of the variables were: external chest: pre-test = 8.0%; after 15 min: 8.8%; after 2 h: 7.5%; after 24 h: 5.5%; after 48 h: 9.5%. Clavicular pectoralis: pre-test = 8.4%; after 15 min: 11.5%; after 2 h: 9.4%; after 24 h: 11.5%; after 48 h: 12.6%. Triceps: pre-test = 13.0%; after 15 min: 5.0%; after 2 h: 8.2%; after 24 h: 9.7%; after 48 h: 10.4%. Deltoid: pre-test = 6.9%; after 15 min: 6.8%; after 2 h: 6.9%; after 24 h: 9.7%; after 48 h: 8.4%.

The ICC between the recovery methods (passive, DN, and CWI) for each muscle was: external chest: pre-test: ICC = 0.10 (95% CI: −0.24; 0.63, *p* = 0.2); after 15 min: ICC = 0.09 (95% CI: −0.24; 0.62, *p* = 0.3); after 2 h: ICC = 0.13 (95% CI: −0.22; 0.65, *p* = 0.2); after 24 h: ICC = −0.25 (95% CI: −0.41; 0.20, *p* = 0.8); after 48 h: ICC = 0.00 (95% CI: −0.29; 0.54, *p* = 0.4). Clavicular sinus: pre-test: ICC = 0.12 (95% CI: −0.23; 0.64, *p* = 0.2); after 15 min: ICC = 0.39 (95% CI: −0.02; 0.80, *p* = 0.03); after 2 h: ICC = 0.15 (95% CI: −0.21; 0.66, *p* = 0.2); after 24 h: ICC = 0.02 (95% CI: −0.28; 0.56, *p* = 0.4); after 48 h: ICC = 0.525 (95% CI: 0.10; 0.86, *p* = 0.007). Triceps: pre-test: ICC = −0.01 (95% CI −0.30; 0.53, *p* = 0.4); after 15 min: ICC = 0.32 (95% CI: −0.08; 0.77, *p* = 0.06); after 2 h: ICC = 0.24 (95% CI: −0.15; 0.72, *p* = 0.1); after 24 h: ICC = −0.0 (95% CI: −0.34; 0.43, *p* = 0.6); after 48 h: ICC = 0.26 (95% CI: −0.13; 0.73, *p* = 0.1). Deltoid: pre-test: ICC = −0.16 (95% CI: −0.37; 0.35, *p* = 0.7); after 15 min: ICC = −0.32 (95% CI: −0.44; 0.06, *p* = 0.9); after 2 h: ICC = 0.35 (95% CI: −0.06; 0.78, *p* = 0.05); after 24 h: ICC = −0.02 (95% CI: −0.30; 0.52, *p* = 0.5); after 48 h: ICC = 0.04 (95% CI: −0.27; 0.58, *p* = 0.3).

## 4. Discussion

To our knowledge, this study is the first to investigate the effects of different methods of recovery on Paralympic powerlifting athletes. This study aimed to evaluate the effects of three different post-training recovery methods on mechanical, biochemical, muscle edema, and pain indicators in Paralympic powerlifting athletes.

This study’s important results are as follows: (1) Maximal force decreased from pretest to post-test after application of different recovery methods. (2) Only the DN increased IL-2 levels at different time points. (3) After DN, muscle thickness did not increase; however, after CWI, muscle thickness was higher after 15 min and 2 h. (4) Pain pressure threshold reduced after applying the DN method.

The results show that the recovery methods can alter the MIF magnitude. A significant effect of the recovery method was found for MIF. The results revealed that the magnitude of MIF decreased compared to pretest values and the implementation of the different recovery methods. Both PR and DN increased to their respective baseline levels at 2 and 24 h, whereas with cold-water immersion MIF increased from 2 to 24 h but reached 20% more force after 24 h than that at baseline level (Figure 2). It is difficult to explain these results. Some studies have assessed the effectiveness of CWI on muscle stiffness after strenuous exercise [9,10,12,33]. Although CWI reduces muscle pain 24 h after exercise, our study results suggest that CWI does not affect muscle thickness.

There is an increasing body of evidence that exercises, including the strength training exercise, can alter different cytokine plasma levels [34]. IL-2 is a cytokine signaling molecule in the immune system that regulates lymphocyte activities. It is part of the body’s natural response to microbial infection that helps discriminate foreign (“non-self”) cells from “self” cells (31). IL-4 induces differentiation of naive helper T cells to Th2 cells [34]. There has been growing interest among scientists to examine the immune system’s behavioral response to exercise. Therefore, we investigated the role of exercise sessions on IL-2, IL-4, and TNF plasma levels. It was found that the recovery methods had a significant effect only on IL-2 at the tested time points. The results also demonstrated that CWI and DN methods increased IL-2 levels from 24 to 48 h more than that from 2 to 24 h.

Historically, inflammation has been thought to be detrimental for recovery from exercise. However, it is now widely accepted that if tightly regulated, inflammatory responses are integral to muscle repair and regeneration [35]. The authors of different studies revealed that various cell types, including neutrophils, macrophages, mast cells, eosinophils, CD8 and T-regulatory lymphocytes, and adipogenic fibro progenitors pericytes, help facilitate muscle tissue regeneration [36,37]. However, more research is required to determine whether these cells respond to exercise-induced muscle damage. A large body of research has investigated the efficacy of physiotherapeutic, pharmacologic, and nutritional interventions for reducing the signs and symptoms of exercise-induced muscle damage, with mixed results [36,37,38].

Regarding muscle thickness, a significant effect of the recovery method on muscle thickness was noted. This study noticed that the muscle thickness increased mostly after PR in all muscles and remained elevated, whereas only the muscle thickness was significantly higher after 15 min and 2 h after CWI. After the DN method, muscle thickness did not increase significantly in any of the muscles, and after 2h, muscle thickness decreased significantly again in the major pectoralis muscle (Figure 3). Exercise seems to induce a continuous increase in elbow flexors´ thickness from 24 to 96 h after strength training [39]. In our study, both DP and CWI reduced the effects of exercise on muscle thickness.

We also investigated the pain pressure threshold. Exercise sessions seemed to induce an elevation in the pain perception, which was significantly affected by the recovery method used for the pectoralis muscles, not for the deltoid muscles. The results also revealed that the pain pressure threshold increased significantly immediately after the implementation of all recovery methods (15 min). However, the pain pressure threshold continued to decrease for all muscles, with the lowest value occurring 24 h after PR and increasing after that. In the DN recovery method, a similar development as that in PR occurred, but the reduction in pain pressure threshold decreased, in general, less than that in PR. After CWI, pain pressure stabilized after 15 min and significantly increased after 2 h for the pectoralis clavicular part (Figure 4).

An important aspect of our study is that these analyses were performed in a particular group of Paralympic athletes. On the one hand, these data provide new insights into recovery in high-level Paralympic athletes, whereas, on the other hand, interpretation of these data is difficult because of the lack of literature on this topic. Therefore, our study contributes to the area of sports science by shedding light on a particular group of Paralympic powerlifting athletes’ adaptive responses to different recovery methods.

However, despite the relevance of the results, the present study has some limitations: (1) sample size; (2) a short period of experiment; and (3) the athletes’ diet was not controlled during the study period. These limitations can interfere with the recovery process. Therefore, we suggest the inclusion of extra measurements across a prolonged period.

## 5. Conclusions

The strength training session was able to generate changes in metabolism, and different recovery methods contributed differently to the return of homeostasis in Paralympic powerlifting athletes. Muscle damage can be caused by metabolic, structural, and microvascular changes, with systemic and local effects. Because of the above, the results show that interventions alter some aspects of local recovery in edema and pain. However, there are still systemic effects.

It seems that cold water has a good recovery up to 24 and 48 h later and dry needling appears to be a good option for 24 h after the training session. In this sense, the work presents the effective dry needling methods for shorter-term recoveries and CWI presents itself as a good alternative for slightly shorter and longer recoveries.

## Figures and Tables

**Figure 1 ijerph-18-05155-f001:**
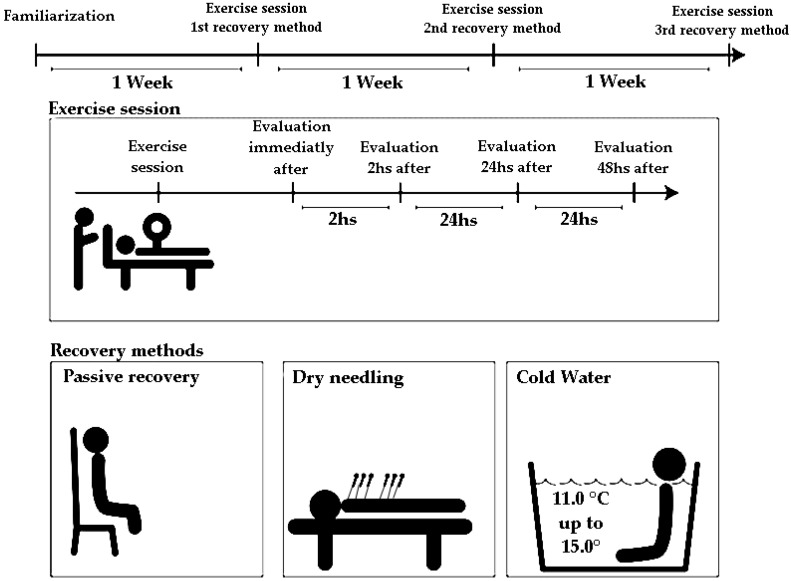
Outline of the research: summary of the study. The athletes in three types of recovery methods: passive, dry needling, and cryotherapy. The recovery methods were evaluated in four stages: immediately, 2 h, 24 h, and 48 h after the end of the session.

**Figure 2 ijerph-18-05155-f002:**
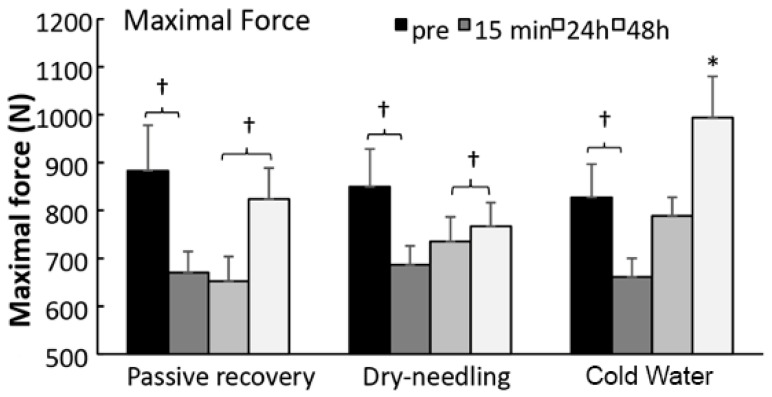
Maximal isometric force measured in the bench press before and after 15 min, 2 h, 24 h, and 48 h after an exercise induces muscle damage and a recovery (passive, dry-needling, or cold-water immersion) protocol after 15 min, 24 h, and 48 h after exercise-induced muscle damage and a recovery protocol. * indicates a significant difference with all other test moments (*p* < 0.05). † indicates a significant difference between these two test moments (*p* < 0.05).

**Figure 3 ijerph-18-05155-f003:**
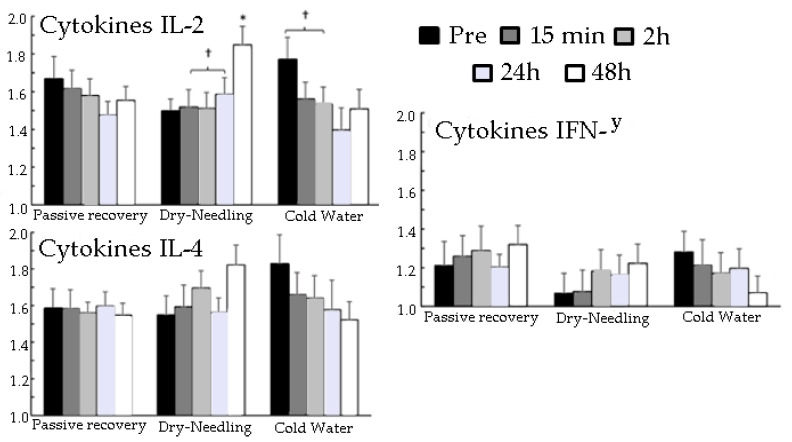
Mean (±SEM) differences in blood biochemical indicators (Cytokines IL-2, IL-4, and IFN-γ). Before and after 15 min, 2 h, 24 h, and 48 h exercise-induced muscle damage and recovery (passive, dry-needling, or cold-water immersion) protocol. Cytokines IL-2, IL-4 e, and IFN-γ (pg/mL). * indicates a significant difference with all other test occasions (*p* < 0.05). † indicates a significant difference between these two test occasions (*p* < 0.05).

**Figure 4 ijerph-18-05155-f004:**
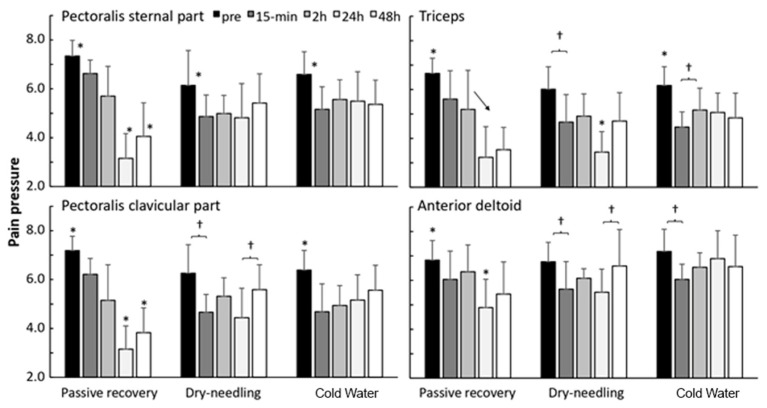
Mean (±SEM) differences in pain pressure before and after 15 min, 2 h, 24 h, and 48 h exercise-induced muscle damage and recovery (passive, dry-needling, or cold-water immersion) protocol for triceps, pectoralis clavicular and sternal part, and anterior deltoid muscles. Pressure values (kg/cm^2^). * indicates a significant difference with all other test occasions (*p* < 0.05). † indicates a significant difference between these two test occasions. (*p* < 0.05) indicates a significant change between these test moments for this protocol on a *p* < 0.05 level.

**Figure 5 ijerph-18-05155-f005:**
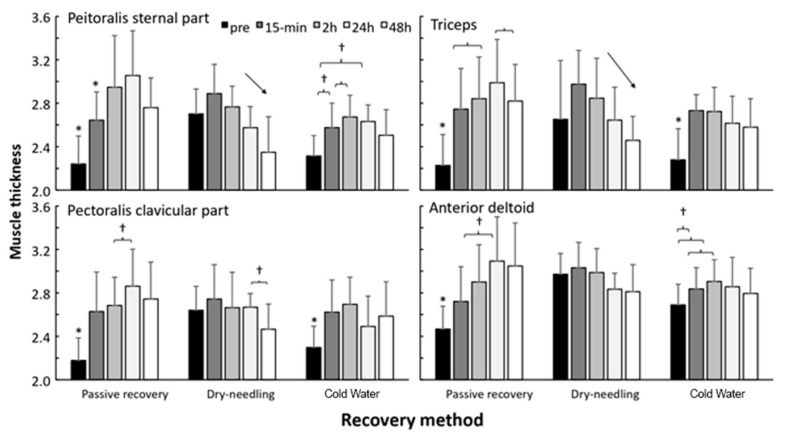
Mean (±SEM) differences in muscle thickness measured with ultrasound before and after 15 min, 2 h, 24, and 48 h exercise-induced muscle damage and recovery (passive, dry-needling, or cold-water immersion) protocol for triceps, pectoralis clavicular, sternal part, and anterior deltoid muscles. Muscle thickness (cm). * indicates a significant difference with all other test occasions (*p* < 0.05). † indicates a significant difference between these two test occasions. (*p* < 0.05) indicates a significant change between these test moments for this protocol on a *p* < 0.05 level.

## Data Availability

The data that support this study can be obtained from the address www.ufs.br/GPEPS, Accessed on 20 February 2021.

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
