# Peer review of "Physiological and Biochemical Evaluation of Different Types of Recovery in National Level Paralympic Powerlifting"

_ijerph, 2021, doi:10.3390/ijerph18105155_

Round 1
Reviewer 1 Report
- The authors examined Physiological and biochemical evaluation of different types of 2 recovery in national level Paralympic powerlifting.
- There are some grammatical issues throughout the text that need to be fixed.
Abstract
- Lines 39-40: Participants’ gender needs to be specified in the abstract.
- Line 44: The acronym “IL-2” needs to be used in its complete form first.
Introduction
- Lines 80-81: You need to provide a reference(s) to support the following sentence:
“In addition, the use of dry needling (DN) in recovery has been proposed to reduce muscle pain.”
Methods
- Day-to-day test reliability, CV range, and intraclass correlation coefficients for the assessments need to be included for ALL the assessments.
Discussion
- Lines 317-318: More references are needed to support the following sentence:
“Some studies have assessed the effectiveness of CWI on muscle stiffness after strenuous exercise [33].”
Author Response
We thank you for your notes and the proposed adaptations are attached

Reviewer 2 Report
The article ijerph-1211730 entitled "Physiological and biochemical evaluation of different types of recovery in national level Paralympic powerlifting", seems to me an interesting and novel article and I congratulate the authors for it.
However, I think that some minimal modifications should be taken care of:
- Authors should indicate whether the study subjects are male or female.
- They should clearly indicate what the inclusion and exclusion criteria were in the choice of subjects.
- In relation to the dry puncture method, indicate how they sterilized the surface where they were punctured and in which parts of the muscle they were punctured.
- In the Blood biochemical indicators section, it should be included how the serum was obtained.
- In line 206 of the text it indicates 200 rpm, they are sure, because I think there are few revolutions for a phase separation.
For the rest, confirm my congratulations on the article made and on the group made.
Author Response

(The authors gave the same response as above.)
